# Uncertainty Modeling for Out-of-Distribution Generalization

**Xiaotong Li[1], Yongxing Dai[1], Yixiao Ge[2], Jun Liu[3], Ying Shan[2], Ling-Yu Duan[1,4]***
[1]Peking University, Beijing, China    [2]ARC Lab, Tencent PCG
[3]Singapore University of Technology and Design, Singapore
[4]Peng Cheng Laboratory, Shenzhen, China
`lixiaotong@stu.pku.edu.cn, {yongxingdai, lingyu}@pku.edu.cn,`
`{yixiaoge, yingsshan}@tencent.com, jun_liu@sutd.edu.sg`

## Abstract

Though remarkable progress has been achieved in various vision tasks, deep neural networks still suffer obvious performance degradation when tested in out-of-distribution scenarios. We argue that the feature statistics (mean and standard deviation), which carry the domain characteristics of the training data, can be properly manipulated to improve the generalization ability of deep learning models. Common methods often consider the feature statistics as deterministic values measured from the learned features and do not explicitly consider the uncertain statistics discrepancy caused by potential domain shifts during testing. In this paper, we improve the network generalization ability by modeling the uncertainty of domain shifts with synthesized feature statistics during training. Specifically, we hypothesize that the feature statistic, after considering the potential uncertainties, follows a multivariate Gaussian distribution. Hence, each feature statistic is no longer a deterministic value, but a probabilistic point with diverse distribution possibilities. With the uncertain feature statistics, the models can be trained to alleviate the domain perturbations and achieve better robustness against potential domain shifts. Our method can be readily integrated into networks without additional parameters. Extensive experiments demonstrate that our proposed method consistently improves the network generalization ability on multiple vision tasks, including image classification, semantic segmentation, and instance retrieval. The code can be available at https://github.com/lixiaotong97/DSU.

## 1 Introduction

Deep neural networks have shown impressive success in computer vision, but with a severe reliance on the assumption that the training and testing domains follow an independent and identical distribution (Ben-David et al., 2010; Vapnik, 1992). This assumption, however, does not hold in many real-world applications. For instance, when employing segmentation models trained on sunny days for rainy and foggy environments (Choi et al., 2021), or recognizing art paintings with models that trained on photographs (Li et al., 2017), inevitable performance drop can often be observed in such out-of-distribution deployment scenarios. Therefore, the problem of domain generalization, aiming to improve the robustness of the network on various unseen testing domains, becomes quite important.

Previous works (Huang & Belongie, 2017; Li et al., 2021) demonstrate that feature statistics (mean and standard deviation), as the moments of the learned features, carry informative domain characteristics of the training data. Domain characteristics primarily refer to the information that is more specific to the individual domains but less relevant to the task objectives, such as the photo style and capturing environment information in object recognition. Consequently, domains with different data distributions generally have inconsistent feature statistics (Wang et al., 2020b; 2019a; Gao et al., 2021a). Most deep learning methods follow Empirical Risk Minimization principle (Vapnik, 1999) to minimize their average error over the training data (Shen et al., 2021). Despite the satisfactory

---

*Corresponding Author.

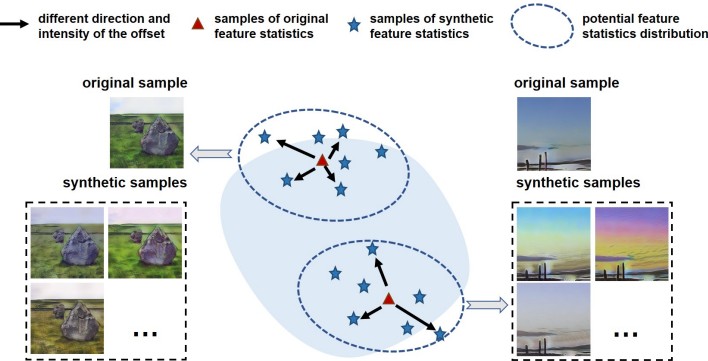

Figure 1: The visualization of reconstructed samples with synthesized feature statistics, using a pre-trained style transfer auto-encoder (Huang & Belongie, 2017). The illustration of the feature statistics shifts, which may vary in both intensity and direction (*i.e.*, different offsets in the vector space of feature statistics). We also show images of "new" domains generated by manipulating feature statistic shifts with different direction and intensity. Note these images are for visualization only, rather than feeding into the network for training.

performance on the training domain, these methods do not explicitly consider the uncertain statistics discrepancy caused by potential domain shifts during testing. As a result, the trained models tend to overfit the training domain and show vulnerability to the statistic changes at testing time, substantially limiting the generalization ability of the learned representations.

Intuitively, the test domains may bring uncertain statistics shifts with different potential directions and intensities compared to the training domain (as shown in Figure 1), implying the uncertain nature of domain shifts. Considering such "uncertainty" of potential domain shifts, synthesizing novel feature statistics variants to model diverse domain shifts can improve the robustness of the trained network to different testing distributions. Towards this end, we introduce a novel probabilistic method to improve the network generalization ability by properly *modeling **D**omain **S**hifts with **U**ncertainty* (DSU), *i.e.*, characterizing the feature statistics as uncertain distributions.

In our method, instead of treating each feature statistic as a deterministic point measured from the feature, we hypothesize that the feature statistic, after considering potential uncertainties, follows a multi-variate Gaussian distribution. The distribution "center" is set as each feature's original statistic value, and the distribution "scope" represents the variant intensity considering underlying domain shifts. Uncertainty estimation is adopted here to depict the distribution "scope" of probabilistic feature statistics. Specifically, we estimate the distribution "scope" based on the variances of the mini-batch statistics in an efficient non-parametric manner. Subsequently, feature statistics variants are randomly sampled from the estimated Gaussian distribution and then used to replace the original deterministic values for modeling diverse domain shifts, as illustrated in Figure 2. Due to the generated feature statistics with diverse distribution possibilities, the models can be trained to properly alleviate the domain perturbations and encode better domain-invariant features.

Our proposed method is simple yet fairly effective to alleviate performance drop caused by domain shifts, and can be readily integrated into existing networks without bringing additional model parameters or loss constraints. Comprehensive experiments on a wide range of vision tasks demonstrate the superiority of our proposed method, indicating that introducing uncertainty to feature statistics can well improve models' generalization against domain shifts.

## 2 RELATED WORK

### 2.1 DOMAIN GENERALIZATION

Domain generalization (DG) has been attracting increasing attention in the past few years, which aims to achieve out-of-distribution generalization on unseen target domains using only single or multiple source domain data for training (Blanchard et al. (2011)). Research on addressing this problem has been extensively conducted in the literature (Zhou et al. (2021a); Wang et al. (2021); Shen et al. (2021)). Here some studies that are more related to our work are introduced below.

**Data Augmentation**: Data augmentation is an effective manner for improving generalization ability and relieving models from overfitting in training domains. Most augmentation methods adopt various transformations at the image level, such as AugMix (Hendrycks et al. (2020)) and CutMix (Yun et al. (2019)). Besides using handcraft transformations, mixup (Zhang et al. (2018)) trains the model by using pair-wise linearly interpolated samples in both the image and label spaces. Manifold Mixup (Verma et al. (2019)) further adopts this linear interpolation from image level to feature level. Some recent works extend the above transformations to feature statistics for improving model generalization. MixStyle (Zhou et al. (2021b)) adopts linear interpolation on feature statistics of two instances to generate synthesized samples. The pAdaIn (Nuriel et al. (2021)) swaps statistics between the samples applied with a random permutation of the batch.

**Invariant Representation Learning**: The main idea of invariant representation learning is to enable models to learn features that are invariant to domain shifts. Domain alignment-based approaches (Li et al. (2018c;b)) learn invariant features by minimizing the distances between different distributions. Instead of enforcing the entire features to be invariant, disentangled feature learning approaches (Chattopadhyay et al. (2020); Piratla et al. (2020)) decouple the features into domain-specific and domain-invariant parts and learn their representations simultaneously. In addition, normalization-based methods (Pan et al. (2018); Choi et al. (2021)) can also be used to remove the style information to obtain invariant representations.

**Learning Strategies**: There are also some effective learning strategies that can be leveraged to improve generalization ability. Ensemble learning is an effective technique in boosting model performance. The ensemble predictions using a collection of diverse models (Zhou et al. (2020b)) or modules (Seo et al. (2020)) can be adopted to improve generalization and robustness. Meta-learning-based methods (Finn et al. (2017); Li et al. (2018a); Dai et al. (2021)) learn to simulate the domain shifts following an episode training paradigm. Besides, self-challenging methods, such as RSC (Huang et al. (2020)), force the model to learn a general representation by discarding dominant features activated on the training data.

## 2.2 UNCERTAINTY IN DEEP LEARNING

Uncertainty capturing the "noise" and "randomness" inherent in the data has received increasing attention in deep representation learning. Variational Auto-encoder (Kingma & Welling (2013)), as an important method for learning generative models, can be regarded as a method to model the data uncertainty in the hidden space. Dropout (Srivastava et al. (2014)), which is widely used in many deep learning models to avoid over-fitting, can be interpreted to represent model uncertainty as a Bayesian approximation (Gal & Ghahramani (2016)). In some works, uncertainty is used to address the issues of low-quality training data. In person re-identification, DistributionNet (Gal & Ghahramani (2016)) adopts uncertainty to model the person images of noise-labels and outliers. In face recognition, DUL (Chang et al. (2020)) and PFE ((Shi & Jain, 2019)) apply data uncertainty to simultaneously learn the feature embedding and its uncertainty, where the uncertainty is learned through a learnable subnetwork to describe the quality of the image. Different from the aforementioned works, our proposed method is used to model the feature statistics uncertainty under potential domain shifts and acts as a feature augmentation method for handling our-of-distribution generalization problem.

## 3 METHOD

### 3.1 PRELIMINARIES

Given $x \in \mathbb{R}^{B \times C \times H \times W}$ to be the encoded features in the intermediate layers of the network, we denote $\mu \in \mathbb{R}^{B \times C}$ and $\sigma \in \mathbb{R}^{B \times C}$ as the channel-wise feature mean and standard deviation of each instance in a mini-batch, respectively, which can be formulated as:

$$\mu(x) = \frac{1}{HW} \sum_{h=1}^{H} \sum_{w=1}^{W} x_{b,c,h,w}, \tag{1}$$

$$\sigma^2(x) = \frac{1}{HW} \sum_{h=1}^{H} \sum_{w=1}^{W} (x_{b,c,h,w} - \mu(x))^2. \tag{2}$$

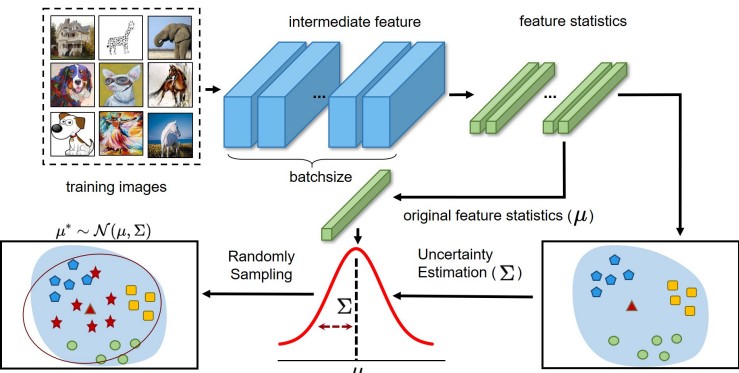

Figure 2: Illustration of the proposed method. Feature statistic is assumed to follow a multi-variate Gaussian distribution during training. When passed through this module, the new feature statistics randomly drawn from the corresponding distribution will replace the original ones to model the diverse domain shifts.

As the abstraction of features, feature statistics can capture informative characteristics of the corresponding domain (such as color, texture, and contrast), according to previous works (Huang & Belongie, 2017; Li et al., 2021). In out-of-distribution scenarios, the feature statistics often show inconsistency with training domain due to different domain characteristics (Wang et al., 2019a; Gao et al., 2021a), which is ill-suited to deep learning modules like nonlinearity layers and normalization layer and degenerates the model's generalization ability (Wang et al., 2020b). However, most of the deep learning methods only treat feature statistics as deterministic values measured from the features while lacking explicit consideration of the potential uncertain statistical discrepancy. Owing to the model's inherent vulnerability to such discrepancy, the generalization ability of the learned representations is limited. Some recent methods (Nuriel et al., 2021; Zhou et al., 2021b) utilize feature statistics to tackle the domain generalization problem. Despite the success, they typically adopt linear manipulation (*i.e.*, exchange and interpolation) on pairwise samples to generate new feature statistics, which limits the diversity of synthetic changes. Specifically, the direction of their variants is determined by the chosen reference sample and such internal operation restricts their variant intensity. Thus these methods are sub-optimal when handling the diverse and uncertain domain shifts in real world.

## 3.2 MODELING DOMAIN SHIFTS WITH UNCERTAINTY

Given the arbitrary testing domains with uncertain feature statistic shifts in both direction and intensity, properly modeling the domain shifts becomes an important task for tackling the challenge of domain generalization problem.

Considering the uncertainty and randomness of domain shifts, it is promising to employ the methods of "uncertainty" to treat the "uncertainty" of domain shifts. In this paper, we propose a novel method by *modeling **D**omain **S**hifts with **U**ncertainty* (DSU). Instead of treating each feature statistic as a deterministic value measured from the learned feature, we hypothesize that the distribution of each feature statistic, after considering potential uncertainties, follows a multi-variate Gaussian distribution. This means each feature statistic has a probabilistic representation drawn from a certain distribution, *i.e.*, the feature statistics mean and standard deviation follow $\mathcal{N}(\mu, \Sigma_\mu^2)$ and $\mathcal{N}(\sigma, \Sigma_\sigma^2)$, respectively. Specifically, the corresponding Gaussian distribution's center is set as each feature's original statistics, while the Gaussian distribution's standard deviation describes the uncertainty scope for different potential shifts. Through randomly sampling diverse synthesized feature statistics with the probabilistic approach, the models can be trained to improve the robustness of the network against statistics shifts.

### 3.2.1 UNCERTAINTY ESTIMATION

Taking the uncertainty of domain shifts into consideration, the uncertainty estimation in our method aims to depict the uncertainty scope of each probabilistic feature statistic. However, the testing domain is unknown, which makes it challenging to obtain an appropriate variant range.

Some generative-based studies (Shen & Zhou, 2021; Wang et al., 2019b) show that the variances between features contain implicit semantic meaning and the directions with larger variances can imply potentials of more valuable semantic changes. Inspired by this, we propose a simple yet effective non-parametric method for uncertainty estimation, utilizing the variance of the feature statistics to provide some instructions:

$$\Sigma_\mu^2(x) = \frac{1}{B} \sum_{b=1}^{B} (\mu(x) - \mathbb{E}_b[\mu(x)])^2, \tag{3}$$

$$\Sigma_\sigma^2(x) = \frac{1}{B} \sum_{b=1}^{B} (\sigma(x) - \mathbb{E}_b[\sigma(x)])^2. \tag{4}$$

where $\Sigma_\mu \in \mathbb{R}^C$ and $\Sigma_\sigma \in \mathbb{R}^C$ represent the uncertainty estimation of the feature mean $\mu$ and feature standard deviation $\sigma$, respectively. The magnitudes of uncertainty estimation can reveal the possibility that the corresponding channel may change potentially. Although the underlying distribution of the domain shifts is unpredictable, the uncertainty estimation captured from the mini-batch can provide an appropriate and meaningful variation range for each feature channel, which does not harm model training but can simulate diverse potential shifts.

### 3.2.2 PROBABILISTIC DISTRIBUTION OF FEATURE STATISTICS

Once the uncertainty estimation of each feature channel is obtained, the Gaussian distribution for probabilistic feature statistics can be established. To use randomness to model the uncertainty, we adopt the random sampling to further exploit the uncertainty in the probabilistic representations. The new feature statistics, mean $\beta(x) \sim \mathcal{N}(\mu, \Sigma_\mu^2)$ and standard deviation $\gamma(x) \sim \mathcal{N}(\sigma, \Sigma_\sigma^2)$, can be randomly drawn from the corresponding distributions as:

$$\beta(x) = \mu(x) + \epsilon_\mu \Sigma_\mu(x), \qquad \epsilon_\mu \sim \mathcal{N}(\mathbf{0}, \mathbf{1}), \tag{5}$$

$$\gamma(x) = \sigma(x) + \epsilon_\sigma \Sigma_\sigma(x), \qquad \epsilon_\sigma \sim \mathcal{N}(\mathbf{0}, \mathbf{1}). \tag{6}$$

Here we use the re-parameterization trick (Kingma & Welling (2013)) to make the sampling operation differentiable, and $\epsilon_\mu$ and $\epsilon_\sigma$ both follow the standard Gaussian distribution. By exploiting the given Gaussian distribution, random sampling can generate various new feature statistics information with different combinations of directions and intensities.

### 3.2.3 IMPLEMENTATION

The implementation of our method is by the means of AdaIN (Huang & Belongie (2017)), and replaces the feature statistics with the randomly drawing ones to achieve the transformation. The final form of the proposed method can be formulated as:

$$\text{DSU}(x) = \underbrace{(\sigma(x) + \epsilon_\sigma \Sigma_\sigma(x))}_{\gamma(x)} \left( \frac{x - \mu(x)}{\sigma(x)} \right) + \underbrace{(\mu(x) + \epsilon_\mu \Sigma_\mu(x))}_{\beta(x)}. \tag{7}$$

The above operation can be integrated at various positions of the network as a flexible module. Note that the module only works during model training and can be discarded while testing. To trade off the strength of this module, we set a hyperparameter $p$ that denotes the probability to apply it. The algorithm is described in the Appendix. Benefiting from the proposed method, the model trained with uncertain feature statistics will gain better robustness against potential statistics shifts, and thus acquires a better generalization ability.

# 4 EXPERIMENTS

In order to verify the effectiveness of the proposed method in improving the generalization ability of networks, we conduct the experiments on a wide range of tasks, including image classification, semantic segmentation, instance retrieval, and robustness towards corruptions, where the training and testing sets have different cases of distribution shifts, such as style shift, synthetic-to-real gap, scenes change, and pixel-level corruption.

## 4.1 GENERALIZATION ON MULTI-DOMAIN CLASSIFICATION

**Setup and Implementation Details:** We evaluate the proposed method on PACS (Li et al. (2017)), a widely-used benchmark for domain generalization with four different styles: Art Painting, Cartoon, Photo, and Sketch. The implementation follows the official setup of MixStyle (Zhou et al. (2021b)) with a *leave-one-domain-out* protocol and ResNet18 (He et al., 2016) is used as the backbone. The random shuffle version of MixStyle is adopted for fair comparisons, which does not use domain labels. In addition to PACS, we also employ Office-Home (Venkateswara et al., 2017) for multi-domain generalization experiments in the Appendix.

**Experiment Results:** The experiments results, shown in Table 1, demonstrate our significant improvement over the baseline method, which shows our superiority to the conventional deterministic approach. Especially in Art and Sketch, our method has nearly 10% improvement in average accuracy. Furthermore, our method also outperforms the competing methods, which indicates our method that models diverse uncertain shifts on feature statistics is effective to improve network generalization ability against different domain shifts. Photo has similiar domain characteristics as ImageNet dataset and the slight drop might be due to the ImageNet pretraining (also discussed in (Xu et al., 2021)). Our DSU augments the features and enlarges the diversity of the training domains. In contrast, the baseline method preserves more pre-trained knowledge from ImageNet thus tends to overfit the Photo style dataset benefiting from pretraining.

Table 1: Experiment results of PACS multi-domain classification task. RSC* denotes the reproduced results from pAdaIN (Nuriel et al., 2021).

| Method | Reference | Art | Cartoon | Photo | Sketch | Average (%) |
|---|---|---|---|---|---|---|
| Baseline | - | 74.3 | 76.7 | **96.4** | 68.7 | 79.0 |
| Mixup (Zhang et al., 2018) | ICLR 2018 | 76.8 | 74.9 | 95.8 | 66.6 | 78.5 |
| Manifold Mixup (Verma et al., 2019) | ICML 2019 | 75.6 | 70.1 | 93.5 | 65.4 | 76.2 |
| CutMix (Yun et al., 2019) | ICCV 2019 | 74.6 | 71.8 | 95.6 | 65.3 | 76.8 |
| RSC* (Huang et al., 2020) | ECCV 2020 | 78.9 | 76.9 | 94.1 | 76.8 | 81.7 |
| L2A-OT (Zhou et al., 2020a) | ECCV 2020 | 83.3 | 78.2 | 96.2 | 73.6 | 82.8 |
| SagNet (Nam et al., 2021) | CVPR 2021 | 83.6 | 77.7 | 95.5 | 76.3 | 83.3 |
| pAdaIN (Nuriel et al., 2021) | CVPR 2021 | 81.7 | 76.6 | 96.3 | 75.1 | 82.5 |
| MixStyle (Zhou et al., 2021b) | ICLR 2021 | 82.3 | 79.0 | 96.3 | 73.8 | 82.8 |
| DSU | Ours | **83.6** | **79.6** | 95.8 | **77.6** | **84.1** |

## 4.2 GENERALIZATION ON SEMANTIC SEGMENTATION

**Setup and Implementation Details:** Semantic segmentation, as a fundamental application for automatic driving, encounters severe performance declines due to scenarios differences (Wang et al., 2020a). GTA5 (Richter et al., 2016) is a synthetic dataset generated from Grand Theft Auto 5 game engine, while Cityscapes (Cordts et al., 2016) is a real-world dataset collected from different cities in primarily Germany. To evaluate the cross-scenario generalization ability of segmentation models, we adopt synthetic GTA5 for training while using real CityScapes for testing. The experiments are conducted on FADA released codes (Wang et al. (2020a)), using DeepLab-v2 (Chen et al., 2018) segmentation network with ResNet101 backbone. Mean Intersection over Union (mIOU) and mean Accuracy (mAcc) of all object categories are used for evaluation.

Table 2: Experiment results of semantic segmentation from synthetic GAT5 to real Cityscapes.

| Method | Reference | mIOU (%) | mAcc (%) |
|---|---|---|---|
| Baseline | - | 37.0 | 51.5 |
| pAdaIN (Nuriel et al., 2021) | CVPR 2021 | 38.3 | 52.1 |
| Mixstyle (Zhou et al., 2021b) | ICLR 2021 | 40.3 | 53.8 |
| DSU | Ours | **43.1** | **57.0** |

**Experiment Results:** Table 2 shows the experiment results compared to related methods. As for a pixel-level classification task, improper changes of feature statistics might constrain the performances. The variants generated from our method are centered on the original feature statistics with different perturbations. These changes of feature statistics are mild for preserving the detailed information in these dense tasks. Meanwhile, our method can take full use of the diverse driving scenes and generates diverse variants, thus show a significant improvement on mIOU and mAc by 6.1% and 5.5%, respectively. The visualization result is shown in Figure 3.

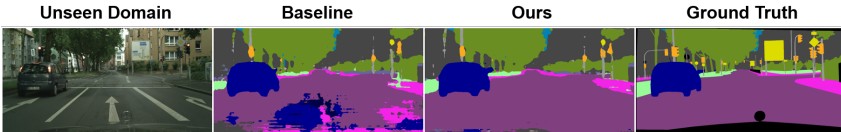

Figure 3: The visualization on unseen domain Cityscapes with the model trained on synthetic GTA5.

### 4.3 GENERALIZATION ON INSTANCE RETRIEVAL

**Setup and Implementation Details:** In this section, person re-identification (ReID), which aims at matching the same person across disjoint camera views, is used to verify the effectiveness of our method on the instance retrieval task. Experiments are conducted on the widely used DukeMTMC (Ristani et al. (2016)) and Market1501 (Zheng et al. (2015)) datasets. The implementation is based on MMT (Ge et al., 2020) released codes and ResNet50 is adopted as the backbone. Meanwhile, mean Average Precision (mAP) and Rank-1 (R1) precision are used as the evaluation criterions.

Table 3: Experiment results of instance retrieval on ReID dataset DukeMTMC and Market1501. A → B denotes models are trained on A while evaluated on B. For fair comparisons, we reproduce the experiments under the same framework.

| Method | Reference | Market → Duke | | Duke → Market | |
|---|---|---|---|---|---|
| | | mAP (%) | R1 (%) | mAP (%) | R1 (%) |
| Baseline | - | 25.8 | 42.3 | 26.7 | 54.7 |
| pAdaIN (Nuriel et al., 2021) | CVPR 2021 | 28.0 | 46.1 | 27.9 | 56.1 |
| MixStyle (Zhou et al., 2021b) | ICLR 2021 | 28.2 | 46.7 | 28.1 | 56.6 |
| DSU | Ours | **32.0** | **52.0** | **32.4** | **63.7** |

**Experiment Results:** ReID is a fine-grained instance retrieval task, where the subtle information of persons is important for retrieving an instance. MixStyle and pAdain rely on a reference sample to generate new feature statistics, which might introduce confounded information from the reference sample. Compared to them, our method does better in maintaining the original information and also has more variant possibilities. The experiment results are demonstrated in Table 3. Our method achieves huge improvement compared to the baseline method and also outperform MixStyle and pAdaIN by a big margin.

### 4.4 ROBUSTNESS TOWARDS CORRUPTIONS

**Setup and Implementation Details:** We validate the proposed method for robustness towards corruptions on ImageNet-C (Hendrycks & Dietterich (2019)), which contains 15 different pixel-level corruptions. ResNet50 is trained with 100 epochs for convergence on large-scale ImageNet-1K (Deng et al. (2009)) and the hyperparameter $p$ is set as $0.1$ for training in ImageNet. We also add our method on APR (Chen et al. (2021)), a recently state-of-the-art method on ImageNet-C, to verify that our method can be compatible with other image-level augmentation methods. Error is adopted as the evaluation metric for clean ImageNet. Mean Corruption Error (mCE) is adopted as evaluation metric for ImageNet-C, which is computed as the average of the 15 different corruption errors and normalized by the corruption error of AlexNet (Krizhevsky et al. (2012)).

**Experiment Results:** Although the corruptions are imposed on the pixel level, they still introduce a shift in the statistics (Benz et al., 2021). So our method shows consistent improvement on ImageNet-C. Meanwhile, the instances in the testing set may not always fall into the distribution of the training

Table 4: Experiment results of clean image classification on ImageNet, and the robustness toward corruptions on ImageNet-C.

| | Clean (↓) | Corrupted (↓) | Noise | | | Blur | | | | Weather | | | | Digital | | | |
|---|---|---|---|---|---|---|---|---|---|---|---|---|---|---|---|---|---|
| | Error (%) | mCE (%) | Gauss | Shot | Implulse | Defocus | Glass | Motion | Zoom | Snow | Frost | Fog | Bright | Contrast | Elastic | Pixel | JPEG |
| Baseline | 23.8 | 76.2 | 80 | 81 | 83 | 75 | 87 | 76 | 80 | 78 | 74 | 67 | 56 | 70 | 83 | 76 | 73 |
| DSU | **23.4** | **73.4** | 76 | 77 | 78 | 71 | 83 | 77 | 79 | 74 | 71 | 66 | 55 | 68 | 82 | 65 | 71 |
| APR | 24.0 | 65.0 | 52 | 56 | 50 | 69 | 85 | 69 | 79 | 62 | 64 | 55 | 54 | 63 | 84 | 65 | 65 |
| APR+DSU | **23.7** | **64.1** | 51 | 56 | 49 | 69 | 84 | 67 | 78 | 61 | 63 | 51 | 53 | 56 | 83 | 66 | 65 |

set, and they still have slight statistic shifts (Gao et al. (2021b)). Thus it can be seen that the within-dataset ImageNet accuracy is also increased. When combining APR with our method, mCE can be decreased from 65.0 % to 64.1%, showing that our method can be compatible with state-of-the-art methods on ImageNet-C for further improvement.

## 5 ABLATION STUDY

In this section, we perform an extensive ablation study of the proposed method on PACS and segmentation task (GTA5 to Cityscapes) with models trained on ResNet. The effects of different inserted positions and hyper-parameter of the proposed method are analyzed below. Meantime, we also analyze the effects on different choices of uncertainty distribution.

**Effects of Different Inserted Positions:** DSU can be a plug-and-play module to be readily inserted at any position. Here we name the positions of ResNet after first Conv, Max Pooling

Table 5: Effects of different inserted positions.

| Inserted Positions | Baseline | 0-3 | 1-4 | 2-5 | 0-5 |
|---|---|---|---|---|---|
| PACS | 79.0 | 82.2 | 83.1 | 83.5 | 84.1 |
| GTA5 to Cityscapes | 37.0 | 41.1 | 40.9 | 42.1 | 43.1 |

layer, 1,2,3,4-th ConvBlock as 0,1,2,3,4,5 respectively. As shown in Table 5, no matter where the modules are inserted, the performances are consistently higher than the baseline method. The results show that inserting the modules at positions 0-5 would have better performances, which also indicates modeling the uncertainty in all training stages will have better effects. Based on the analysis, we plug the module into positions 0-5 in all experiments.

**Effects of Hyper-parameter:** The hyper-parameter of the probability $p$ is to trade off the strength of feature statistics augmentation. As shown in Figure 4, the results are not sensitive to the probability setting and the accuracy reaches the best results when setting $p$ as 0.5, which is also adopted as the default setting in all experiments if not specified.

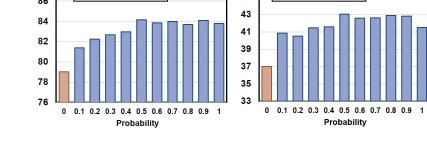

Figure 4: The effects on the hyper-parameter probability.

**Choices of Uncertainty Distribution**: In our method, the Gaussian distribution with uncertainty estimation is adopted as the default setting, we also conduct other distributions for comparisons in Table 6. Specifically, Random denotes directly adding random shifts draw from a fixed Gaussian $\mathcal{N}(0, 1)$, and Uniform denotes that the shifts are drawn from $U(-\Sigma, \Sigma)$, where $\Sigma$ is the scope obtained from our uncertainty estimation. As we can see, directly using Gaussian distribution with the improper variant scope will harm the model performances, indicating the variant range of feature statistics should have some instructions. Further analysis about different vanilla Gaussian distributions with pre-defined standard deviation are conducted in the Appendix. Meanwhile, the result of Uniform shows some improvement but is still lower than DSU, which indicates the boundless Gaussian distribution is more helpful to model more diverse variants.

Table 6: Different choices of distribution for uncertainty.

| Choice | Baseline | Random | Uniform | DSU |
|---|---|---|---|---|
| PACS | 79.0 | 76.9 | 81.9 | 84.1 |
| GTA5 to Cityscapes | 37.0 | 38.2 | 41.6 | 43.1 |

# 6 FURTHER ANALYSIS

## 6.1 QUANTITATIVE ANALYSIS ON THE PROPOSED METHOD

In this subsection, we will analyze the effects of the proposed method on both intermediate features and feature representations. Quantitative experiments are conducted on PACS, where we choose Art Painting as the unseen testing domain and the rests are used as training domains.

To study the phenomena of feature statistic shifts, we capture the intermediate features after the second block in ResNet18 and measure the average feature statistics values of one category in the training and testing domain, respectively. The distributions of feature statistics are shown in Figure 5. As the previous works (Wang et al., 2020b; 2019a) show, the feature statistics extracted from the baseline model show an obvious shift due to different data distribution. It can be seen that the model trained with our method has less shift. Our method can help the model gain robustness towards domain shifts, as it properly models the potential feature statistic shifts.

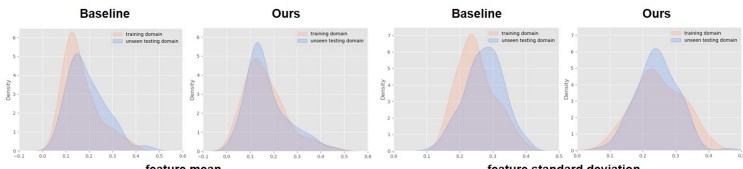

Figure 5: Quantitative analysis on the shifts of feature statistics (mean and standard deviation) between training source domains and unseen testing domain.

## 6.2 VISUALIZATION ON THE SYNTHETIC CHANGES

Besides the quantitative experiment results, we also obtain a more intuitional view of the diverse changes provided by our method, through visualizing the reconstruction results using a predefined autoencoder[1] (Huang & Belongie (2017)), where the proposed module is inserted into the encoder, and inverse the feature representations into synthetic images after the decoder. As the results shown in Figure 6, the reconstructed images obtained from our probabilistic approach show diverse synthetic changes, such as the environment, object texture, and contrast, etc.

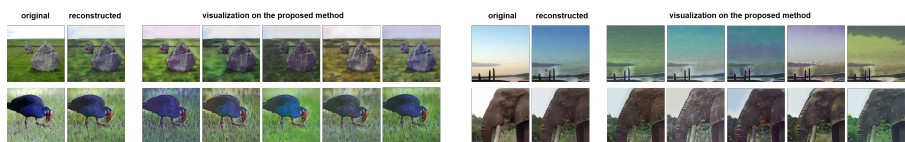

Figure 6: The visualization on diverse synthetic changes obtained from our method.

# 7 CONCLUSIONS

In this paper, we propose a probabilistic approach to improve the network generalization ability by modeling the uncertainty of domain shifts with synthesized feature statistics during training. Each feature statistic is hypothesized to follow a multi-variate Gaussian distribution for modeling the diverse potential shifts. Due to the generated feature statistics with diverse distribution possibilities, the models can gain better robustness towards diverse domain shifts. Experiment results demonstrate the effectiveness of our method in improving the network generalization ability.

ACKNOWLEDGEMENT

This work was supported by the National Natural Science Foundation of China under Grant 62088102, and in part by the PKU-NTU Joint Research Institute (JRI) sponsored by a donation from the Ng Teng Fong Charitable Foundation.

---

[1]The pretrained weights are from https://github.com/naoto0804/pytorch-AdaIN.

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

## A  APPENDIX

### A.1  ALGORITHM

The algorithm of the proposed method is illustrated in Algorithm 1.

---

**Algorithm 1:** The algorithm of the proposed method (DSU)

---

**Input:** Intermediate feature $x \in \mathbb{R}^{B \times C \times H \times W}$, probability $p$ to forward this module;
**Output:** Intermediate feature $\widehat{x} \in \mathbb{R}^{B \times C \times H \times W}$ after considering potential statistics shifts;

1  Sample $p_0 \sim U(0, 1)$;
2  **if** $p_0 < p$ **and Training then**
3      Compute the channel-wise mean and standard deviation of each instance in a mini-batch;
4      $\mu(x) = \frac{1}{HW} \sum\limits_{h=1}^{H} \sum\limits_{w=1}^{W} x_{b,c,h,w}$,
5      $\sigma^2(x) = \frac{1}{HW} \sum\limits_{h=1}^{H} \sum\limits_{w=1}^{W} (x_{b,c,h,w} - \mu(x))^2$.
6      Uncertainty estimation on feature statistics;
7      $\Sigma_\mu^2(x) = \frac{1}{B} \sum\limits_{b=1}^{B} (\mu_{bc}(x) - E_b(\mu_{bc}(x)))^2$,
8      $\Sigma_\sigma^2(x) = \frac{1}{B} \sum\limits_{b=1}^{B} (\sigma_{bc}(x) - E_b(\sigma_{bc}(x)))^2$.
9      Compute the synthetic feature statistics randomly sampling from the given Guassian distributions;
10     $\beta(x) = \mu(x) + \epsilon_\mu \Sigma_\mu(x), \quad \epsilon_\mu \sim \mathcal{N}(\mathbf{0}, \mathbf{1})$,
11     $\gamma(x) = \sigma(x) + \epsilon_\sigma \Sigma_\sigma(x), \quad \epsilon_\sigma \sim \mathcal{N}(\mathbf{0}, \mathbf{1})$.
12     Obtain the feature after considering potential statistics shifts;
13     $\widehat{x} = \gamma(x) \times \frac{x - \mu(x)}{\sigma(x)} + \beta(x)$.
14     return the feature $\widehat{x}$ with uncertain feature statistics.
15 **else**
16     adopt the original feature $x$ and skip this module.
17 **end**

---

### A.2  MULTI-DOMAIN GENERALIZATION ON OFFICE HOME.

In addition to multi-domain classification experiments on PACS, we further evaluate the effectiveness of the proposed method on Office-Home (Venkateswara et al., 2017), which contains 15,500 images of 65 classes for home and office recognition. The experiment results with ResNet18 backbone are shown in Table 7. It can be observed that our method brings obvious improvement over the baseline method and also outperforms the competing methods. By introducing the feature statistics uncertainty, the models trained with our method can learn to alleviate the domain perturbations, such as the style information, and obtain more domain-invariant features. For example, huge improvement can be observed from the results on Clipart, which is a domain with much different style from others.

Table 7: Experiment results of Office-Home multi-domain classification task.

| Method | Reference | Art | Clipart | Product | Real | Average (%) |
|---|---|---|---|---|---|---|
| Baseline | - | 58.8 | 48.3 | 74.2 | 76.2 | 64.4 |
| Mixup (Zhang et al., 2018) | ICLR 2018 | 58.2 | 49.3 | 74.7 | 76.1 | 64.6 |
| CrossGrad (Shankar et al., 2018) | ICLR 2018 | 58.4 | 49.4 | 73.9 | 75.8 | 64.4 |
| Manifold Mixup (Verma et al., 2019) | ICML 2019 | 56.2 | 46.3 | 73.6 | 75.2 | 62.8 |
| CutMix (Yun et al., 2019) | ICCV 2019 | 57.9 | 48.3 | 74.5 | 75.6 | 64.1 |
| RSC (Huang et al., 2020) | ECCV 2020 | 58.4 | 47.9 | 71.6 | 74.5 | 63.1 |
| L2A-OT (Zhou et al., 2020a) | ECCV 2020 | 60.6 | 50.1 | 74.8 | 77.0 | 65.6 |
| MixStyle (Zhou et al., 2021b) | ICLR 2021 | 58.7 | 53.4 | 74.2 | 75.9 | 65.5 |
| DSU | Ours | 60.2 | 54.8 | 74.1 | 75.1 | 66.1 |

### A.3 Choice of uncertainty distribution

Table 8: Intensive study about different vanilla Gaussian distributions with pre-defined standard deviation.

| Choice | Baseline | Rand($10^0$) | Rand($10^{-1}$) | Rand($10^{-2}$) | Rand($10^{-3}$) | DSU |
|---|---|---|---|---|---|---|
| PACS | 79.0 | 76.9 | 81.2 | 79.3 | 79.1 | 84.1 |
| GTA5 to Cityscapes | 37.0 | 38.2 | 39.8 | 40.1 | 38.9 | 43.1 |

Besides the analysis of the uncertainty estimation in the ablation study, we also conduct a more intensive study about the effects of pre-defined uncertainty estimations. Specifically, Rand($s$) denotes directly imposing random shifts draw from a fixed Gaussian $\mathcal{N}(0, s^2)$. The intensive study is shown in Table 8. It can be observed that the results of different fixed distributions are all much lower than the proposed method. Some conclusions could be obtained from the results. (a): Imposing excessive uncertainty might harm the model training and degrade the performance. (b): The best fixed value of uncertainty estimation might vary from different tasks. By contrast, the proposed method can be adaptive to different tasks without any manual adjustment.

Table 9: Study about the effects of sharing the same uncertain distribution among different channels.

| Choice | Baseline | Channel-share | DSU |
|---|---|---|---|
| PACS | 79.0 | 80.2 | 84.1 |
| GTA5 to Cityscapes | 37.0 | 39.3 | 43.1 |

We also conduct the experiment to test the effectiveness of treating different channels with different potentials. Channel-share denotes all channels of the sample share the same uncertainty distribution, *i.e.,* using the average uncertainty estimation among channels. As shown in Table 9, the results indicate that sharing the same uncertain distribution among different channels is less effective, which ignores the different potentials of channels and will limit their performances. Meanwhile, the proposed method explicitly considers the different potentials of different channels and brings better performances.

### A.4 T-SNE visualization

To analyze the effects on feature representations, we visualize the feature representation vectors of different categories in unseen domain with t-SNE (Van der Maaten & Hinton, 2008) in Figure 7. The features of the same category become more compact benefiting from the proposed method. Because our method can alleviate the domain perturbations during training and make the model focus on content information, obtaining more invariant features representations.

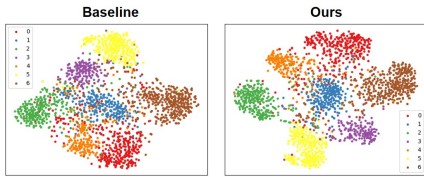

Figure 7: The t-SNE visualization on unseen PACS domain.

### A.5 Comparisons to the related methods

Some related methods (Zhou et al., 2021b; Nuriel et al., 2021) also tackle the domain generalization problem by producing synthetic feature statistics. Specifically, we denote the random shuffle copies of the batch feature as $\hat{x} = \text{shuffle}(x)$. pAdaIN (Nuriel et al. (2021)) generate new samples by swapping feature statistics between the batch samples applied with a random permutation, where $\beta(x) = \mu(\hat{x})$ and $\gamma(x) = \sigma(\hat{x})$. MixStyle (Zhou et al. (2021b)) generates synthesized domain samples by mixing feature statistics information of two instances, where $\beta(x) = \lambda\mu(x) + (1 - \lambda)\mu(\hat{x})$ and $\gamma(x) = \lambda\sigma(x) + (1 - \lambda)\sigma(\hat{x})$ and $\lambda \in (0, 1)$ is a random interpolation weight.

Despite the success, they typically adopt linear manipulation on pairwise samples to generate new feature statistics, which limits the diversity of synthetic changes. Specifically, the direction of the variants is determined by the chosen reference sample and the internal operation also restricts the variant intensity. Our method, not relies on a specific reference sample, is based on the Gaussian distribution that can produce not only linear changes but diverse variants with more possibilities. Due to the boundless range of the Gaussian distribution, our method has the ability to generate feature statistics beyond the scope of training domain, which also breaks the limitation of inner interpolation between training samples. The visualization of the comparisons is shown in Figure 8.

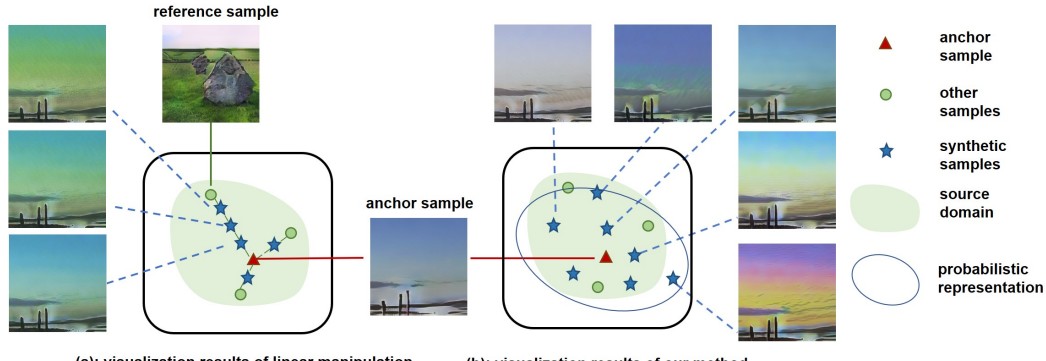

Figure 8: Comparisons with related methods. The variants produced by previous pairwise-based methods are restricted by the combination of chosen sample pair, while our method can generate various feature statistics variants with different combination of directions and intensities.

## A.6  WITHIN-DATASET PERFORMANCE

In Table 4, we tested the within-dataset performance on the large-scale dataset ImageNet, denoted as "Clean". We observed that the top-1 error rate declines from 23.8% to 23.4% after training with the proposed DSU, indicating that DSU does not sacrifice the in-domain performance to gain the benefits on out-of-distribution domains.The reason might be that instances in the testing set may not always fall into exactly the same distribution of the training set, and they still have slight statistic shifts (Gao et al., 2021a). The proposed DSU can help the trained model improve the robustness to statistics shifts and thus gain better performance in within-dataset ImageNet.

Besides the experiments on ImageNet, we also supplement the within-dataset performances on PACS. According to the multi-source training protocol on PACS (Li et al., 2017), the within-domain performance is averaged over multiple training-domain datasets (P,A,C,S denotes Art, Cartoon, Photo and Sketch respectively). As shown in Table 10, it can be observed that the within-dataset performance of our DSU also slightly beats the baseline, verifying the conclusion as on ImageNet.

Table 10: Within-dataset performance on PACS. P,A,C,S denote Photo, Art Painting, Cartoon, and Sketch respectively.

| Method | Reference | P,C,S | P,A,S | A,C,S | P,A,C | Average (%) |
|--------|-----------|-------|-------|-------|-------|-------------|
| Baseline | - | 95.70 | 95.28 | 94.22 | 96.58 | 95.44 |
| DSU | Ours | 96.20 | 96.32 | 95.17 | 97.20 | 96.21 |

## A.7  ABLATION STUDY ON BATCH SIZE

In Table 11, we conduct an ablation study on the batch size. As shown in the table, consistent performance gains are observed with various batch sizes on PACS. Note we use the batch size of 64 in our paper, following the original setting in PACS (Li et al., 2017) for fair comparison.

Table 11: Ablation study on the effects of batch size.

| batchsize | Reference | 16 | 32 | 64 | 128 | 256 |
|-----------|-----------|-----|-----|-----|------|------|
| Baseline | - | 81.0 | 80.2 | 79.0 | 77.8 | 75.6 |
| DSU | Ours | 84.9 (+3.9) | 84.5 (+4.3) | 84.1 (+5.1) | 82.1 (+4.3) | 80.4 (+4.8) |

