# OpenReview forum: "Uncertainty Modeling for Out-of-Distribution Generalization"
_ICLR.cc/2022/Conference — ICLR 2022 Poster_

### Official Review · Reviewer_jq8e · 2021-11-01

**Correctness:** 4
**Technical Novelty And Significance:** 3
**Empirical Novelty And Significance:** 3
**Recommendation:** 6
**Confidence:** 4

**Main Review:**

Strengths:

(1) The proposed method can capture the uncertainty of the feature domain, which is very important to address the domain shift issue.

(2) The proposed method can achieve some improvement on cross-dataset evaluation for multiple vision tasks compared with other domain generalization methods.

(3) This paper is well written and well-organized.

Weaknesses:

For the method:

(1) The proposed method is simple and is similar to batch normalization. Can traditional batch normalization method also improve model generalization ability? Is the proposed method significantly better than traditional batch normalization?

(2) Generating the class-preserving augmented features is important since we do not want the generated features to be far away from the original class domain. The proposed feature augmentation method did not consider the class-preserving issue, i.e., how to guarantee the augmented feature is still located within the feature domain of the same class.

(3) We may expect more discussion on why the domain generalization can be improved by modeling the probabilistic distribution of features. Is there a specific theory or reason?

For the experiments:

(1) The authors can provide some within-dataset experiments for comparison.

(2) According to Eq.(3) and Eq.(4), the variance of feature statistics is related to batches, it is better to perform an ablation study of the influence of batch size.

(3) For the image classification problem, the performance of DSU drops on Photo style compared with the baseline. Are there some explanations?



**Summary Of The Paper:**

This paper proposed a feature augmentation method for domain generalization by generating probabilistic feature statistics, where the distribution of feature statistics enlarges the feature domain with possible domain shifts. The proposed method is evaluated on various vision tasks, including image classification, semantic segmentation, and instance retrieval. It achieves competitive results compared with recent domain generalization methods.

**Summary Of The Review:**

Overall, this paper is well-written and well-organized for both theories and experiments. The idea is interesting and is easy to implement. The experiment results are empirically sound.

---

> ### Author Response · Authors · 2021-11-18
> **Responses to Reviewer jq8e (1/3)**
>
> Thank you for the constructive and positive comments! We really appreciate the comments for improving the clarity of statements and experimental verifications. The manuscript is revised accordingly (marked in blue), and the main concerns are listed as below.
>
> **Q1:**
>
> (1) The proposed method is simple and is similar to batch normalization.
>
> (2) Can traditional batch normalization method also improve model generalization ability? Is the proposed method significantly better than traditional batch normalization?
>
> **A1:**
> (1) The proposed method explicitly models the uncertainty of domain shifts with probabilistic feature statistics, sampled from the Gaussian distribution via uncertainty estimation, which provides a simple and general feature augmentation method for out-of-distribution generalization. Despite the simplicity in formulation, a wide range of experiments show the effectiveness of our method.
>
> As for the comparison between our DSU and batch normalization (BN), though both exploiting the feature statistics, they are very different. Specifically, BN acts as a technic to normalize the features to a standard distribution for faster and more stable training of deep neural networks. In contrast, our DSU acts as a feature augmentation method to improve the model generalization ability by generating probabilistic feature statistics to model the diverse domain shifts.
>
> (2) BN improves *in-domain* test performances because it well normalizes features using the statistics from the training data (domain). However, BN is less effective when testing in *out-of-distribution* domains due to the statistics inconsistency [1,2]. It is because the statistics moving averaged from the *training* domain (i.e., running mean and variance) are often not suitable for *testing* in out-of-distribution domains, which will limit the model's generalization ability.
>
> In contrast, our DSU properly models the statistics inconsistency with probabilistic feature statistics during training, resulting in much better performances in out-of-distribution scenarios. Note that DSU is only used during training in order to make the training model to capture more domain possibilities, which is discarded in testing and does not need to modify the original architectures.
>
> The experimental results below show the significant performance gains of our DSU compared to BN in different tasks. Note that the result of BN is denoted as the baseline in our paper, i.e., a vanilla backbone without extra techniques for improving the model generalization ability.
>
> |     | PACS (Acc) | GTA5 to CityScapes (mAcc)| Duke to Market (Rank1)|
> |  :-:  | :-:  | :-:| :-: |
> | BN (baseline)  | 79.0% |51.5%|54.7%|
> | Ours  | 84.1%|57.0% |63.7%|
>
> References:
>
> [1] Wang, Ximei, et al. "Transferable normalization: Towards improving transferability of deep neural networks." NIPS. 2019.
>
> [2] Wang, Ximei, et al. "Transferable calibration with lower bias and variance in domain adaptation." NIPS. 2020.
>
>
> **Q2:** Generating the class-preserving augmented features is important since we do not want the generated features to be far away from the original class domain. The proposed feature augmentation method did not consider the class-preserving issue, i.e., how to guarantee the augmented feature is still located within the feature domain of the same class.
>
> **A2:**
> As shown in previous studys [3,4], the feature statistics reflect more on the domain characteristics (also referred to as style), thus the class information can still be well preserved after transformation on feature statistics. The reconstructed visualization in Figure 7 also well supports this claim, i.e., the contents including the classes are preserved after the proposed feature transformation on statistics.
>
> We are also glad to conduct the ablation study by modeling the uncertainty of the feature statistics only within samples of each class. As shown in the table below, inferior performance can be observed compared to our original design. It is actually due to the fact that class-wise augmentation limits the diversity of synthesized domain distributions, resulting in the sub-optimal generalization ability of the trained model.
>
> |   PACS  | Art Painting | Cartoon| Photo| Sketch| Avg|
> |  :-:  | :-:  | :-:| :-: | :-:| :-: |
> |Baseline|74.3%|76.7%|96.4%|68.7%|79.0%|
> |Class-wise|82.2%|79.7%|95.4%|74.9%|83.0%|
> |Ours|83.6%|79.6%|95.8%|77.6%|84.1%|
>
> References:
>
> [3] Huang, Xun, and Serge Belongie. "Arbitrary style transfer in real-time with adaptive instance normalization." ICCV. 2017.
>
> [4] Chandran, Prashanth, et al. "Adaptive Convolutions for Structure-Aware Style Transfer." CVPR. 2021.

---

> > ### Author Response · Authors · 2021-11-18
> > **Responses to Reviewer jq8e (2/3)**
> >
> > **Q3:** We may expect more discussion on why the domain generalization can be improved by modeling the probabilistic distribution of features. Is there a specific theory or reason?
> >
> > **A3:**
> > Previous works [1,2,5,6] demonstrate that domains with different data distributions often have inconsistent feature statistics. The feature statistics inconsistency is ill-suited to deep learning modules like nonlinearity layers and normalization layers [1,2,6], which will cause cascaded errors during model inference and degenerates the model's generalization ability [5]. Our motivation is to consider the potential statistics shifts and explicitly model the uncertainty of the feature domain to improve the model robustness towards potential shifts. As shown in Figure 5, in our paper, the model trained with our method shows less statistics shift as expected on out-of-distribution testing, which demonstrates the effectiveness of modeling the probabilistic distribution of features. We have revised the paper to make it more clear (marked in blue).
> >
> >
> > References:
> >
> > [1] Wang, Ximei, et al. "Transferable normalization: Towards improving transferability of deep neural networks." NIPS. 2019.
> >
> > [2] Wang, Ximei, et al. "Transferable calibration with lower bias and variance in domain adaptation." NIPS. 2020.
> >
> > [5] Gao, Shang-Hua, et al. "Representative batch normalization with feature calibration." CVPR. 2021.
> >
> > [6] Luo, Ping, et al. "Towards understanding regularization in batch normalization." ICLR. 2018.
> >
> > **Q4:** The authors can provide some within-dataset experiments for comparison.
> >
> > **A4:** We tested the within-dataset performance on the large-scale dataset ImageNet in Table 4, denoted as "Clean". We observed that the top-1 error rate declines from 23.8% to 23.4% after training with the proposed DSU, indicating that DSU does not sacrifice the in-domain performance to gain the benefits on out-of-distribution domains. The reason might be that instances in the testing set may not always fall into exactly the same distribution of the training set, and they still have slight statistic shifts [5]. The proposed DSU can help the trained model improve the robustness to statistics shifts and thus gain better performance in within-dataset ImageNet.
> >
> > Besides the experiments on ImageNet, we also supplement the within-dataset performances on PACS. According to the multi-source training protocol on PACS [7], the within-domain performance is averaged over multiple training-domain datasets (P,A,C,S denote Art, Cartoon, Photo, and Sketch respectively). As shown in the table below, it can be observed that the within-dataset performance of our DSU also outperforms the baseline, verifying the same conclusion as on ImageNet. We have added the above comparisons in the revised paper, referring to Table 10.
> >
> > |     | P,C,S | P,A,S | A,C,S | P,A,C| Avg|
> > |  :-:  | :-:  | :-:| :-: | :-:| :-: |
> > |Baseline|95.70%|95.28%|94.22%|96.58%|95.44%|
> > |Ours|96.20%|96.32%|95.17%|97.20%|96.21%|
> >
> > Reference:
> >
> > [7] Li, Da, et al. "Deeper, broader and artier domain generalization." ICCV. 2017.

---

> > > ### Author Response · Authors · 2021-11-18
> > > **Responses to Reviewer jq8e (3/3)**
> > >
> > > **Q5:**  According to Eq.(3) and Eq.(4), the variance of feature statistics is related to batches, it is better to perform an ablation study of the influence of batch size.
> > >
> > > **A5:** We follow your suggestions and conduct such an ablation study on the batch size. As shown in the table below, consistent performance gains are observed with various batch sizes on PACS. Note we use the batch size of 64 in our paper, following the original setting in PACS [7] for fair comparison. We have added the ablation study in the revised paper in Table 11.
> > >
> > > |  batchsize   | 16|32|64|128|256|
> > > |-|-|-|-|-|-|
> > > |Baseline|81.0%|80.2%|79.0%|77.8%|75.6%|
> > > |Ours|84.9% (+3.9%)|84.5% (+4.3%)|84.1% (+5.1%)|82.1% (+4.3%)|80.4% (+4.8%)|
> > >
> > > Reference:
> > >
> > > [7] Li, Da, et al. "Deeper, broader and artier domain generalization." ICCV. 2017.
> > >
> > > **Q6:** For the image classification problem, the performance of DSU drops on Photo style compared with the baseline. Are there some explanations?
> > >
> > > **A6:**
> > > As observed in many state-of-the-art works [8,9,10], they also show similar drops on Photo dataset. A possible reason of the drops might be the ImageNet pretraining, which is also discussed in [8]. Photo is a small-scale dataset (7 classes, 3929 samples in total), which has very similar domain characteristics to the dataset ImageNet (1000 classes, 128k samples) for pre-training. Our method aims to augment features and enlarge the diversity of training domains (Art, Cartoon, and Sketch style). In contrast, the baseline method may retain more knowledge from ImageNet and thus tends to overfit the Photo dataset benefiting from pretraining. As a result, the baseline shows slightly better performance than other competing methods and our method on the Photo dataset.
> > > Overall, our DSU gains +5.1% on average precision, +9.3% on Art, and +8.9% on Sketch compared to baseline in PACS, which demonstrates our effectiveness. Thanks for the valuable suggestion, we have added the explanations in the revised paper (marked in blue).
> > >
> > > References:
> > >
> > > [8] Xu, Zhenlin, et al. "Robust and generalizable visual representation learning via random convolutions." ICLR. 2020.
> > >
> > > [9] Li, Da, et al. "Episodic training for domain generalization." ICCV. 2019.
> > >
> > > [10] Chattopadhyay, Prithvijit, et al. "Learning to balance specificity and invariance for in and out of domain generalization." ECCV, 2020.

---

> > > > ### Comment · Reviewer_jq8e · 2021-11-21
> > > > **Responces to the Authors**
> > > >
> > > > Thanks for the authors’ responses. The authors conducted multiple experiments that address all my concerns. Overall, I think it is a good work and I will keep my original score of 6.

---

### Official Review · Reviewer_SDxo · 2021-11-02

**Correctness:** 4
**Technical Novelty And Significance:** 4
**Empirical Novelty And Significance:** 4
**Recommendation:** 8
**Confidence:** 3

**Main Review:**

1. Overall, the paper is well written and nicely organized.

2. The proposed method is intuitive and simple. It adds randomness towards the feature statistics to handle out-of-distribution testing. The randomness is also drawn from the observed uncertainty estimations from data, with the hope that the uncertainty estimations from mini-batch could provide variation range to cover the possible unknown domain shifts. The method is simple yet very effective according to the experimental results. This is a big plus for the method.

3. The method is tested on various benchmarks, and shows strong performance compared to recent state-of-the-art methods.

4. Some parts of the paper feel repetitive. For example, Section 3.1, related work and introduction have similar contents, which could be reorganized to improve the presentation and writing.

**Summary Of The Paper:**

This paper studies the problem of out-of-distribution generalization by modeling uncertainty in feature statistics. It improves the network generalization ability by modeling the uncertainty of domain shifts with synthesized feature statistics during training. Instead of being deterministic values, the feature statistics are hypothesized to follow a multivariate Gaussian distribution. The proposed method is tested on various tasks including image classification, semantic segmentation, and instance retrieval, and shows strong performance compared to state-of-the-art methods.

**Summary Of The Review:**

In summary, the method is simple yet effective, showing strong performance on various benchmarks.

---

> ### Author Response · Authors · 2021-11-18
> **Responses to Reviewer SDxo**
>
> Thank you for the positive and helpful comments!
>
> **Q1:** Some parts of the paper feel repetitive. For example, Section 3.1, related work and introduction have similar contents, which could be reorganized to improve the presentation and writing.
>
> **A1:** We really appreciate the comments for improving the presentation and writing, and the manuscript is revised accordingly (marked in blue). The main revisons are listed below: (1) in Section 3.1, the repetitive parts with introduction has been rewritten for more concise expression. (2) the advantages of our method discussed in related work 2.2 have been merged in introduction. (3) the analysis of MixStyle and pAdaIN in related work 2.1 are merged in Section 3.1 and further comparisons are conducted on Apendix A.4.

---

### Official Review · Reviewer_xdeV · 2021-11-02

**Correctness:** 3
**Technical Novelty And Significance:** 2
**Empirical Novelty And Significance:** 2
**Recommendation:** 6
**Confidence:** 3

**Main Review:**

Strength:
1. The proposed multivariate Gaussian based approach makes sense and technically sound to me.
2. Extensive experiments, ablation study, and sufficient analysis validated the approach empirically.
3. Writing is good and easy to follow.

Weakness:
1. The proposed approach seems to be a small change to AdaIN.
2. Instead of sampling from a Gaussian distribution, can we just normalize it to be an uniform destruction? Similar idea to Batch Normalization, normalizing the distribution to a standard one seems to be an easier choice.

**Summary Of The Paper:**

In this paper, the authors proposed to model uncertainty with multivariate Gaussian distribution for better network generalization. Experiments on multiple benchmark datasets show improved result. Several visualizations and analysis also illustrate the effectiveness of the proposed approach.

**Summary Of The Review:**

Based on the above analysis, I am slightly leaning towards acceptance for now.

---

> ### Author Response · Authors · 2021-11-18
> **Responses to Reviewer xdeV**
>
> Thank you! We really appreciate the positive and valuable comments and the manuscript has been revised (marked in blue). The responses to your main concerns are listed below.
>
> **Q1:** The proposed approach seems to be a small change to AdaIN.
>
> **A1:** The differences between AdaIN and the proposed method mainly fall into two parts.
>
> *Motivation:*
>
> AdaIN is introduced for the task of image style transfer, which adopts the feature statistics of a target style image to achieve the transformation. In contrast, our method tackles the out-of-distribution generalization problem via feature augmentation, which explicitly models the uncertainty of underlying domain shifts using probabilistic feature statistics.
>
> *Contribution:*
>
> AdaIN introduces a generic formulation for feature statistics transformation, which is not claimed as one of our contributions. Instead, how to properly generate the target feature statistics (Eq. (5)&(6)) is the most important in our task.
> Recall that we model the probabilistic feature statistics with diverse distribution possibilities using uncertain Gaussian distributions, which is novel and totally different from AdaIN.  (1) we hypothesize that each feature statistic follows a multi-variate Gaussian distribution after considering potential uncertainties, instead of deterministic values measured from the learned feature. (2) we propose a non-parametric uncertainty estimation method to depict the uncertainty scope of each probabilistic feature statistics. (3) the target feature statistics, centered with the original feature statistics, are randomly sampled from the uncertain Gaussian distributions, instead of directly measured from a specific sample. (4) experiment results on a wide range of tasks demonstrate our superiority to state-of-the-art methods.
>
>
> **Q2**: Instead of sampling from a Gaussian distribution, can we just normalize it to be an uniform destruction? Similar idea to Batch Normalization, normalizing the distribution to a standard one seems to be an easier choice.
>
> **A2**: We further conduct the experiment following your suggestions (denoted as *Normalization*) , which normalizes the distribution of the samples to a standard one. We also compare with Batch Normalization (BN) and the results are shown below.
>
> |   PACS  | Art Painting | Cartoon| Photo| Sketch| Avg|
> |  :-:  | :-:  | :-:| :-: | :-:| :-: |
> |BN |74.3%|76.7%|96.4%|68.7%|79.0%|
> |*Normalization*|79.6%|76.5%|95.2%|71.5%|80.7%|
> |Ours|83.6%|79.6%|95.8%|77.6%|84.1%|
>
> BN improves *in-domain* test performances because it well normalizes features using the statistics from the training data (domain). However, BN is less effective when testing in *out-of-distribution* domains due to the statistics inconsistency [1,2]. It is because the statistics moving averaged from the *training* set (i.e., running mean and variance) are often not suitable for *testing* in out-of-distribution domains thus limits the model performance.
>
>
> *Normalization* adopts the instance statistics to normalize the features and can rectify the feature distribution. Despite some improvement against domain shifts, normalizing all samples to a standard distribution may make features less discriminative and degrade the diversity of feature representation, which limits the model representation ability.
> Compared to the above normalization methods, the proposed DSU explicitly considers the potential shifts and can make the training model capture more diverse domain characteristics, improving the robustness of the models to out-of-distribution domains to a large extent. Benefiting from the proposed uncertainty estimation, our method properly augments the features with the probabilistic feature statistics, which can well preserve the discriminative information of the original features. Moreover, the proposed module is only adopted in training and will be discarded in testing, which does not change the network architecture.
>
> Besides, we also conduct the experiments of a series of uncertain distributions for comparisons, including different uniform and standard distributions, in Table 6 and 8 of our paper, which demonstrates the superiority of our design for improving the model generalization ability.
>
>
> References:
>
> [1] Wang, Ximei, et al. "Transferable normalization: Towards improving transferability of deep neural networks." NIPS. 2019.
>
> [2] Wang, Ximei, et al. "Transferable calibration with lower bias and variance in domain adaptation." NIPS. 2020.

---

> > ### Comment · Reviewer_xdeV · 2021-11-22
> > **Responses to Authors' Responses**
> >
> > Overall, the authors have addressed my main concerns and I am happy to keep my original rate of '6: marginally above the acceptance threshold'.

---

### Decision · Program_Chairs · 2022-01-20

**Decision:**

Accept (Poster)

**Comment:**

The paper considers the important problem of performance degradation under distribution shift and proposes a simple yet effective method to alleviate this problem. They do so by considering feature statistic to be non-deterministic and rather a multivariate Gaussian distribution.  The model can be integrated into networks without additional parameters and experiments show that it works better than BN as well as if the assumed distribution was uniform. The latter was added during rebuttal period.

There were two main concerns regarding distinguishing the work from AdaIN and baseline that were addressed during rebuttal and some parts of the paper were re-written to address repetition.